# Conditional Deletion of AP-2β in the Periocular Mesenchyme of Mice Alters Corneal Epithelial Cell Fate and Stratification

**DOI:** 10.3390/ijms22168730

**Published:** 2021-08-13

**Authors:** Haydn Walker, Aftab Taiyab, Paula Deschamps, Trevor Williams, Judith A. West-Mays

**Affiliations:** 1Health Sciences Centre, McMaster University, 1280 Main St. W, Hamilton, ON L8S 4L8, Canada; walkeh1@mcmaster.ca (H.W.); taiyab@mcmaster.ca (A.T.); deschamp@mcmaster.ca (P.D.); 2Department of Craniofacial Biology, University of Colorado, Mail Stop 8120, RC1-S, Rm L18 11111, 12801 E. 17th Ave, Aurora, CO 80045, USA; TREVOR.WILLIAMS@cuanschutz.edu

**Keywords:** corneal epithelium, AP-2β, periocular mesenchyme, cell fate, stratification, development, stroma-epithelial signaling

## Abstract

The cornea is an anterior eye structure specialized for vision. The corneal endothelium and stroma are derived from the periocular mesenchyme (POM), which originates from neural crest cells (NCCs), while the stratified corneal epithelium develops from the surface ectoderm. Activating protein-2β (AP-2β) is highly expressed in the POM and important for anterior segment development. Using a mouse model in which AP-2β is conditionally deleted in the NCCs (AP-2β NCC KO), we investigated resulting corneal epithelial abnormalities. Through PAS and IHC staining, we observed structural and phenotypic changes to the epithelium associated with AP-2β deletion. In addition to failure of the mutant epithelium to stratify, we also observed that Keratin-12, a marker of the differentiated epithelium, was absent, and Keratin-15, a limbal and conjunctival marker, was expanded across the central epithelium. Transcription factors PAX6 and P63 were not observed to be differentially expressed between WT and mutant. However, growth factor BMP4 was suppressed in the mutant epithelium. Given the non-NCC origin of the epithelium, we hypothesize that the abnormalities in the AP-2β NCC KO mouse result from changes to regulatory signaling from the POM-derived stroma. Our findings suggest that stromal pathways such as Wnt/β-Catenin signaling may regulate BMP4 expression, which influences cell fate and stratification.

## 1. Introduction

The cornea is a highly specialized transparent tissue that constitutes the anterior surface of the eye. In mice, the cornea is comprised of three major layers: an outer stratified squamous (6–7-layer) epithelium, intermediate avascular stroma and an inner endothelial monolayer. The periocular mesenchyme (POM), which is formed predominantly from neural crest cells (NCCs) and to a lesser extent paraxial mesoderm cells, gives rise to the corneal endothelium and stroma [1,2,3]. POM migration occurs at embryonic day (E) 12.5 into the space between the lens and surface ectoderm (SE) [4]. Between E13.5 and 15.5, the POM cells closest to the lens condense into the endothelium, while the others become extracellular matrix (ECM)-secreting keratocytes forming the stroma [4]. Concurrently, the corneal epithelium forms from the anterior SE. However, it remains only 1–2 cell layers until stratification is initiated at the time of eyelid opening at post-embryonic day (P) 12 [4,5]. Bi-directional signaling between the developing POM-derived stroma and SE-derived epithelium is crucial to ensure normal, functional structure and cell fate of the cornea [6]. Major pathways are involved in the interactions between these tissues, including Wnt/β-Catenin, retinoic acid (RA), and TGF-β signaling [7,8,9,10]. Genetic manipulation of these key regulatory pathways between the POM and SE has been shown to result in abnormalities in one or both of the developing and mature corneal stroma and epithelium [7,8,9,10].

The activating protein-2 (AP-2) family of transcription factors, including AP-2α, AP-2β, AP-2γ, AP-2δ and AP-2ε, are involved in ocular development [11,12,13,14,15,16,17,18]. In particular, AP-2β encoded by gene *Tfap2b*, is expressed in the derivatives of the POM, such as the corneal endothelium and stroma [16]. AP-2β is also strongly expressed in other POM-derived tissues such as the trabecular meshwork, ciliary body muscle and iris stroma [13,16,18]. Of importance, AP-2β and AP-2α are also expressed in the embryonic SE that gives rise to the corneal epithelium [16].

AP-2β null mice exhibit eye defects. However, since these mice do not survive postnatally, they do not allow for assessment of corneal pathology that may arise at later stages [19]. Therefore, our lab has used Wnt1Cre transgenic mice to conditionally delete AP-2β in the NCCs which contribute to the POM [16,20]. These mutants, known as AP-2β neural crest knockout (AP-2β NCC KO) mice, exhibit loss of AP-2β expression in the POM and POM-derived tissues, including the corneal stroma and endothelium. Various anterior segment defects were observed in the AP-2β NCC KO mice including a closed iridocorneal angle, corneolenticular adhesions, and significant corneal abnormalities [16]. Our previous studies involving these mutants primarily focused on the glaucomatous changes observed, including elevated intraocular pressure and the progressive loss of retinal ganglion cells. However, detailed investigation of the corneal defects was not performed [16,21].

We have previously noted that the AP-2β NCC KO mutant corneal endothelium was absent, and the stroma was both vascularized and hypercellular [16,21]. Although stromal thickness was similar between the WT and AP-2β NCC KO mice, the mutant stroma was characterized by a less compact structure, including large gaps [16]. Furthermore, the corneal epithelium failed to stratify in the mutants despite the fact that it is of non-POM embryonic origin and AP-2β expression was unaltered in this tissue [16]. In the present study, the corneal epithelial abnormalities in AP-2β NCC KO mice were further explored. We found that in addition to the lack of stratification, cell fate was altered, with central epithelial cells expressing K15, a marker of the limbus and conjunctiva, rather than K12, a definitive marker of the differentiated corneal epithelial phenotype. While we did not find any difference in expression of the key transcription factors PAX6 and P63, we did find downregulation of the growth factor, bone morphogenic protein 4 (BMP4) in the mutant. BMP4 has previously been studied for its role in stromal-epithelial signaling and is potentially of importance in producing the observed epithelial defects observed in the AP-2β NCC KO model.

## 2. Results

While only the corneal stroma and endothelium are derived from the POM, all three corneal tissues in the AP-2β NCC KO mice, including the SE-derived epithelium, displayed abnormalities. As shown with Periodic Acid Schiff (PAS) staining in Figure 1A, at 2–3 months of age, wild-type (WT) mice exhibited all three specialized and distinct corneal tissues with a monolayer endothelium, avascular stroma and stratified epithelium. In contrast, the endothelium was absent in mature AP-2β NCC KO mice and the iris adhered to the corneal tissue. In the central WT epithelium, 6–7 layers of stratified squamous cells with a clear basal columnar layer were seen, posterior to which is the darkly PAS stained epithelial basement membrane (Figure 1A). The peripheral corneal epithelium, approaching the limbus between the cornea and conjunctiva, exhibited fewer stratified layers (Figure 1C). In comparison, the corneal epithelium of the 2–3-month-old AP-2β NCC KO mice was not stratified, being composed of only 1–2 layers of flatter and smaller cells, with no distinct columnar basal cell layer (Figure 1B,D). Both the epithelial and endothelial (Descemet’s membrane) basement membranes were also absent in the AP-2β NCC KO mice compared to WT, as revealed by PAS staining (Figure 1B,D). Adjacent to the corneal epithelium is the conjunctiva, which is not specialized for the passage of light, and includes vasculature and mucus-secreting goblet cells [8,22]. The limbal region between the corneal epithelium and conjunctiva is phenotypically distinct. The corneal limbal epithelial stem cells (LESCs) are found in the basal epithelium for this region and possess proliferative capacity, which allows for maintenance and repair of the corneal epithelium [23,24,25,26].

Epithelial tissues are often differentiated based on the expression of different keratin filaments. In the cornea, mature differentiated epithelial cells can be identified based on expression of the Keratin-12 (K12) protein [27,28,29,30]. The corneal limbal epithelial cells and conjunctival epithelium do not express K12 and instead can be identified by expression of Keratin-15 (K15), a putative marker of LESCs [28,31]. These characteristic markers were used herein to determine the identity of the epithelial cells in the AP-2β NCC KO mice. K12 staining in the 2–3-month-old WT mouse revealed the expected expression pattern, with consistent high levels of K12 across the central cornea, tapering off at the periphery of the epithelium as it approaches the limbus and conjunctiva where no K12 was observed (Figure 2A,C). In the AP-2β NCC KO model K12 expression was completely absent from the entire corneal epithelium (Figure 2B,D). The pattern of K12 expression at earlier developmental timepoints was found to be the same as in the adult WT and mutant mice. As early as P7, K12 was observed in the non-stratified WT epithelium, but not seen in the P7 mutant epithelium (Figure 3). Following eyelid opening, and the initiation of stratification, K12 expression was observed to be further upregulated in the P14 WT compared to the P7 timepoint (Figure 3).

Immunostaining for K15 in the 2–3-month-old WT demonstrated that the central cornea is devoid of expression, instead being upregulated in the epithelium of the conjunctiva and corneal limbal epithelial cells (Figure 4A,C). In comparison, the 2–3-month-old AP-2β NCC KO mice were found to express K15 throughout the central and peripheral corneal epithelium, continuous with expression in the limbus and conjunctiva (Figure 4B,D). Unlike K12 expression, which remained consistent at earlier timepoints, we observed that the K15 pattern in the cornea varies over time. In the unstratified epithelium at P7, K15 staining was observed across the entire extent of the cornea rather than being confined to limbal and conjunctival regions as in the mature cornea (Figure 5). This expression pattern was the same as that observed for the P7 AP-2β NCC KO mice. Following eyelid opening, at P14 we saw that the WT phenotype does match that seen in the mature cornea, with K15 being confined to the limbal region and absent from the rest of the cornea (Figure 5). Similarly, for the mutant at P14, the observed phenotype matched that seen for the 2–3-month-old timepoint, with K15 staining continuing across the full extent of the corneal epithelium (Figure 5).

Based on the observed phenotypic changes to the corneal epithelium, we investigated the expression of transcription factor PAX6, a known regulator of corneal cell fate that has been found to bind directly to the promoter of K12 [32,33]. The Wnt/β-Catenin regulatory pathway has also been shown to negatively regulate PAX6, and as such inhibit the generation of a differentiated epithelial phenotype [34]. Immunostaining for PAX6 in 2–3-month-old WT mice revealed high nuclear expression in all stratified layers, across the central and peripheral cornea, continuing throughout the limbus and conjunctiva (Figure 6A,C). We observed no difference in the AP-2β NCC KO model, with a similar proportion of epithelial cells exhibiting nuclear PAX6 expression across both the central and peripheral cornea, extending through the limbus and conjunctiva (Figure 6B,D). It should be noted that lens epithelial expression of PAX6 was also unchanged in the mutant model relative to the WT, with consistent nuclear expression across the extent of the lens epithelium for both.

Another protein implicated in corneal cell fate determination is BMP4, which is suggested to be necessary for stratification and the promotion of cell differentiation [35,36]. Like PAX6, it has also been shown that BMP4 is negatively regulated by Wnt/β-Catenin signaling [37,38]. Immunostaining for BMP4 in the 2–3-month-old WT mouse revealed that the central corneal layers exhibited consistent basal epithelial expression (Figure 7A). This expression is continuous throughout the peripheral cornea, but tapered off at the limbal region, and no BMP4 expression was observed in the limbal or conjunctival epithelial cells (Figure 7C). A completely altered expression pattern was observed for the mutant, with a complete absence of BMP4 staining across the central and peripheral corneal regions, including the limbus and conjunctiva (Figure 7B,D). Together with the observed staining for K12, these results support the notion that BMP4 plays a role in promoting stratification and differentiation of corneal epithelial cells. From BMP4 IHC staining in the WT at P7 and P14, we also observed upregulation and increased basal epithelial localization of BMP4 post-eyelid opening, coinciding with the initiation of stratification (data not shown).

As a putative LESC marker, P63 was also investigated in the AP-2β NCC KO model to provide insight into the epithelial phenotypic changes and the potential role played by the LESCs [25]. Immunostaining for 2–3-month-old WT mice demonstrated consistent nuclear expression of P63 across the basal epithelial layer of the central and peripheral cornea (Figure 8A). Similar levels of basal epithelial expression were evident in both the limbus and conjunctiva (Figure 8C). Stromal keratocyte expression of P63 was also consistently observed across the central and peripheral regions of the cornea (Figure 8A,C). Staining for P63 in the mutant corneal epithelial showed consistent expression across the central and peripheral cornea, with a similar proportion of cells displaying nuclear staining (Figure 8B,D). Additionally, like the WT, no difference in P63 expression was observed between the limbal and conjunctival epithelia relative to the corneal epithelium. P63 in the stroma of the central and peripheral cornea was also similar the WT (Figure 8B,D). At earlier timepoints, P7 and P14, the P63 expression pattern was not observed to differ between the WT and mutant samples (Figure 9).

## 3. Discussion

Earlier observations of the corneal abnormalities in the AP-2β NCC KO mutant prompted the present study to determine, in greater depth, how the cornea is altered and potential underlying genetic causes. In response to AP-2β deletion in the NCC, it is not surprising that POM-derived tissues including the corneal endothelium and stroma would be impacted. This was observed in the form of an absent endothelium and a hypercellular, vascular stroma [16]. However, it was also found that the SE-derived corneal epithelium was adversely affected, with smaller, flatter cells as well as absent stratification (Figure 1B). Given the non-NCC origin of the epithelium, the defects that we observed suggest that proper epithelial development is contingent on intact expression of AP-2β in the POM-derived stroma.

By examining keratin filament expression, we were able to determine the epithelial identity of the corneal cells and demonstrate that phenotypic changes occurred in the AP-2β NCC KO corneal epithelium. For example, the loss of K12 expression within the mutant corneal epithelium indicates that these cells no longer have the differentiated corneal epithelial cell phenotype (Figure 2) [8,29,30]. This combined with the expansion of K15 expression, a limbal and conjunctival marker, across the extent of the corneal epithelium (Figure 4) provided further evidence of an altered cell fate [28,31]. One possible conclusion to extrapolate from these findings is that a conjunctival-like phenotype has expanded throughout the central corneal epithelial region. In addition to K15 expression, other features of conjunctivalization observed in the mutant included neovascularization of the stroma, loss of the epithelial basement membrane, and absent epithelial stratification [16,21]. However, one prominent feature of conjunctivalization missing in the AP-2β NCC KO corneal epithelial region is the presence of goblet cells [8]. This suggests that the epithelium has not completely shifted to a conjunctival phenotype. As observed in the 2–3-month-old WT littermate mice, K15 is expressed in the conjunctiva, but also in the corneal limbal epithelium (Figure 4). Thus, K15 expression in the central cornea of the mutant may be indicative of an expansion of the basal limbal epithelial cells across the entire cornea. Fitting with this hypothesis is the observation of decreased stratification and an absent basement membrane, neither of which are present in mature WT limbal epithelium (Figure 1). Importantly, goblet cells are not a feature of the limbal epithelium, and as such we would not expect to see them in the central cornea of the AP-2β NCC model if indeed these cells are of limbal cell fate. We plan to investigate conjunctival specific markers and conduct limbal cell fate mapping experiments in our mutant to achieve a greater understanding of phenotypic differences observed as a result of AP-2β deletion in the POM.

PAX6 as a regulator of corneal epithelial cell fate has been shown to bind directly to the promoter of K12, suggesting it is important in controlling the differentiated corneal epithelial cell phenotype [32,33,39]. As such we expected that in the mature WT, PAX6 would be expressed in the regions where we observed K12, and indeed we found PAX6 to be consistently expressed in epithelial nuclei across the cornea, limbus and conjunctiva (Figure 6) in the WT. Interestingly, despite the deficiency of K12 in the AP-2β NCC KO, we did not observe a difference in PAX6 expression as compared to the WT. While PAX6 may be involved in regulation of K12 expression in corneal epithelial cells, the lack of K12 expression observed in the AP-2β NCC KO mutant suggests that other regulators are required. Additional known transcriptions factors that positively regulate K12 include ESE-1 and KLF6 [40,41]. The Kruppel-like factors (KLF) protein family, specifically KLF5 and KLF6, have also been shown to play a role in corneal epithelial development and maintenance including regulating expression of various keratins [42]. Thus, further investigation is required to determine the mechanism by which K12 expression is lost in the AP-2β NCC KO mutant epithelium.

Previous research has shown that BMP4 is an important regulator of corneal stratification as well as involved in determining corneal epithelial cell phenotype [37,38]. We investigated BMP4 to determine if its expression was altered by AP-2β NCC KO. In the 2–3-month-old WT, we observed nuclear BMP4 throughout the central corneal epithelium, though expression was absent from the epithelium of the limbus and conjunctiva (Figure 7). This contrasted sharply with the complete absence of BMP4 staining in the epithelium of the mature mutant cornea. The similar expression patterns for K12 and BMP4 support the notion that BMP4 has a function in determining cell fate. From immunostaining at earlier timepoints (data not shown), we also observed that prior to eyelid opening, the WT and mutant resemble each other closely. However, at P14, after eyelid opening and the initiation of stratification, the WT shows highly specific basal expression in the central cornea. This upregulation of BMP4 in the central epithelium following eyelid opening, as well as its absence in the non-stratified mutant, strongly suggests its role in stratification.

The binding of BMP4 to its receptor complexes initiates a Smad-dependent pathway by which a complex enters the nucleus and regulates transcription factors to control gene expression [43,44]. BMP4 and its receptors are expressed in human and mouse corneal epithelial cells. However, they are also expressed by the stromal keratocytes [35,38]. The process by which BMP4 regulates corneal epithelial development likely involves cross-talk between the mesenchyme and epithelium to regulate necessary gene expression. Various studies have investigated BMP4 in the in the context of its relationship with the Wnt/β-Catenin pathway. In mice, Zhang et al. [38] found evidence that β-Catenin inhibits BMP4 expression in the stroma prior to eyelid opening; however, after eyelid opening, Wnt/β-Catenin signaling is reduced, allowing for upregulation of BMP4 in the stroma [38], which promotes epithelial stratification through Smad-dependent activation of transcription factor P63 [45]. Although these findings focused on paracrine signaling from BMP4 produced in the stroma, the general expression pattern is in agreement with our observation of reduced stratification in the absence of BMP4 for the mutant corneal epithelium (Figure 7) and paralleled our findings of basal epithelial BMP4 expression beginning at P14 (data not shown). Despite stromal BMP4 not being observed in our 2–3-month-old mice, as mentioned previously, the mechanisms of BMP4 signaling are complex and uncertain, and likely involve extensive bi-directional signaling between the stroma and epithelium [36].

As outlined in the results, we observed a distinct expression pattern for BMP4 in the WT, with nuclear expression in the central and peripheral epithelial regions, that was absent in the limbal epithelium (Figure 7). A study from Gouveia et al. using human LESCs found evidence that nuclear β-Catenin suppresses BMP4-mediated cell differentiation as a means of maintaining the stem cell phenotype [37]. In mice, various studies have identified an upregulation of Wnt/β-Catenin signaling at the limbal region, including high levels of the receptor Frizzled-7, providing further support for the notion of Wnt/β-Catenin signaling suppressing BMP4 in the LESCs [8,46,47,48]. Thus, while BMP4 is suppressed in the LESCs by Wnt/β-Catenin signaling, the central cornea exhibits high BMP4 expression, contributing to stratification and differentiation. Based on these findings, and the suppression of BMP4 in the AP-2β NCC KO mice, further investigation of Wnt/β-Catenin signaling in the mutants may provide valuable insight into how corneal epithelial phenotype and stratification are regulated by AP-2β.

Transcription factor P63 is frequently investigated in corneal epithelial research and is hypothesized to play an important role in maintaining proliferative potential and initiating stratification [25,26]. While some studies, often with human tissue, have found evidence that P63 is upregulated in the limbal region as a marker of LESCs [48,49,50], others have observed expression of P63 throughout the basal epithelial cells of the entire cornea in mice [45,51]. In the mature WT mouse, we found that P63 was expressed consistently in basal epithelial cells across the cornea, including both central and limbal regions (Figure 8). In addition, preferential limbal expression of P63 was not observed at earlier timepoints P7 and P14 (Figure 9). As mentioned above, P63 has been proposed as a downstream effector of BMP4 that promotes stratification [38]; however, despite the absence of BMP4 from our mutant, we did not observe any significant change in P63 expression relative to the WT, as was confirmed by cell counts. Across the central and limbal epithelium, a similar proportion of basal epithelial cells displayed nuclear expression of P63 for both the WT and the AP-2β NCC KO model (Figure 8). These findings suggest that the deficient stratification observed as a consequence of AP-2β deletion occurs via a mechanism independent of P63. Despite conflicting evidence in mice, P63 has been shown extensively to be a LESC marker in humans [48,49,50]. Proteins such as YAP and SOX2 have been suggested to regulate P63 in human LESCs in order to maintain proliferative capacity and an undifferentiated state [37,52]. Additional investigation of P63 and its associated regulators, may give further insight into the impact of AP-2β deletion on the LESC population. While previous research did find evidence for BMP4 Smad-dependent regulation of P63, we did not find any change to P63 despite BMP4 being absent in the mutant. BMP4 also initiates downstream Smad-independent signaling, influencing important pathways such as P38 MAPK and ERK1/2 signaling. In our model, BMP4 suppression caused by AP-2β deletion may impact stratification and differentiation through altering Smad-independent effectors of BMP4 signaling rather than P63 through Smad-dependent mechanisms.

A key structural difference between the WT and AP-2β NCC KO corneas was the absence of a PAS-positive corneal epithelial basement membrane in the mutant (Figure 1). In the WT, the basement membrane was observed to be continuous across the central and peripheral cornea, only tapering off at the limbal region. The basement membrane is composed of ECM secreted by the epithelial cells, and, to a lesser extent, the anterior stromal keratocytes. The localization of this structure ensures its involvement with stromal-epithelial bi-directional signaling, and it has been suggested to play an essential role in initiating epithelial stratification [53,54]. Without the basement membrane, the normal regulatory pathways which occur between the stroma and epithelium, including the aforementioned Wnt/β-Catenin pathway, may be impacted and result in changes to cell phenotype or stratification. Adherence between the basement membrane and basal epithelial cells is also essential for bi-directional signaling and involves integrins, a component of hemidesmosomes, binding to their ligands in the basement membrane [55,56,57]. Changes to the composition of the basement membrane or basal cell integrin expression can also impact normal pathways [54,58]. While PAS staining of the basement membrane was absent in the AP-2β NCC KO cornea, which could indicate that the basement membrane has not developed, another possibility is that a basement membrane forms but has a different composition that is not revealed by PAS staining. Additional investigation by our lab (data not shown) has revealed the presence of a basement membrane ultrastructure in the mutant, aligning with the theory that a basement membrane does form, though it possesses a different architecture. An abnormal basement membrane composition could be the result of AP-2β deletion impacting the capacity of the stromal keratocytes or the basal epithelial cells to produce sufficient and proper ECM components for the basement membrane. In turn, this could lead to issues with adhesion to the basal epithelium or impact the ability of regulatory proteins to travel between the stroma and epithelium. Further investigation into how the basement membrane is altered in our mutant could provide insight into changes in the regulatory signaling pathways at large.

## 4. Materials and Methods

### 4.1. Animal Husbandry

All procedures were performed in accordance with the Association for Research in Vision and Ophthalmology (ARVO) Statement for the Use of Animals in Ophthalmic and Vision Research. The AP-2β NCC KO mouse model was investigated with regard to differences in corneal structure and cell phenotype. Previously used by our lab for other studies, the AP-2β NCC KO mice was generated using Cre-LoxP technology. *Tfap2b*, the gene encoding AP-2β, has loxP sites flanking exon 4, and this floxed region was conditionally deleted from the NCC tissue through the use of a Cre transgene that is under the control of the Wnt1 promoter. The mutant was produced through a series of breeding crosses, in which mice hemizygous for the Wnt1Cre transgene, *Wnt1Cre^Tg/0^* (*H2az2^Tg(Wnt1−cre)11Rth^* Tg (Wnt1-GAL2)11Rth/J, Jackson Lab, Bar Harbor, ME, USA), were bred with mice heterozygous for *Tfap2b*, *Tfap2b^+/−^* [16,21]. From this first cross, males found to be *Wnt1Cre^Tg/0^*; *Tfap2b^+/−^* were bred with female mice possessing the loxP flanked *Tfap2b* allele, *Tfap2b^lox/lox^*. Offspring of this cross include the AP-2β NCC KO mice, *Wnt1Cre ^Tg/0^*; *Tfap2b^−/lox^*, as well as littermates possessing a minimum of one active copy of *Tfap2b* which were used as age-matched controls. The desired mutant will therefore have one active copy of *Tfap2b* in all tissues with the exception of the NCCs, where Wnt1 directs Cre recombinase to excise the loxP flanked exon, thus deleting AP-2β expression in the NCC and NCC-derived tissues, including the POM. Necessary genotyping was conducted following standard PCR protocols. Inbreeding between mouse lines used for the final cross was avoided. The background strain used for all breeding crosses was C57BL/6J (Charles River, Wilmington, MA, USA).

### 4.2. Histology

Eyeballs were enucleated from mice euthanized by CO_2_ and fixed in 4% PFA for 2 h prior to washes with 1xPBS and incubation overnight in 70% ethanol. Eyes were then processed and embedded in paraffin wax. Sectioning was conducted at a thickness of 4 μm for both Periodic Acid Schiff (PAS) staining and immunohistochemistry [13].

### 4.3. Immunohistochemistry and Image Analysis

Paraffin-embedded sections were deparaffinized in xylene and then rehydrated in decreasing ethanol concentrations (100%, 95%, and 70% ethanol), before a final wash in water. Heat-mediated antigen retrieval was performed, with sections being placed in 10 mM sodium citrate at pH 6.0 heated to 80–90 °C and maintained at this temperature for 20 min. Having unmasked the protein epitope, an incubation was commenced for 1 h with the normal serum from the host-animal of the secondary antibody, at 5% in 1xPBS to block non-specific staining. A subsequent incubation was conducted overnight at 4 °C with the primary antibody at an appropriate dilution in 1xPBS [13]. The following primary antibodies were used: rabbit anti-K12 (Abcam, Cambridge, UK; ab185627, 1:100), rabbit anti-K15 (Biolegend, San Diego, CA, USA; 833904, 1:100), rabbit anti-PAX6 (Biolegend, San Diego, CA, USA; 901301, 1:100), rabbit anti-BMP4 (Abcam, Cambridge, UK; ab155033, 1:100), and rabbit anti-P63 (Abcam, Cambridge, UK; ab124762, 1:250). After washes with PBS, the appropriate Alexa Fluor secondary antibody, Alexa Fluor 568 (Invitrogen, Molecular Probes, Burlington, CA, USA; 1:200), was added to the sections at a concentration of 1:200 diluted in 1xPBS with 1.5% normal serum from the host. A 1 h incubation time was followed by three 5 min washes with 0.1% Tween-20 solution. Mounting was conducted with ProLong Gold containing DAPI (Invitrogen, Molecular Probes, Burlington, CA, USA) [13]. Imaging was performed using a Leica microscope, with a bright-field attachment for imaging of PAS staining, and fluorescent attachment for immunofluorescence. A high-resolution camera and LasX software (Leica Microsystems, Wetzlar, Germany) were used to acquire images.

A minimum sample size of 3 was used for all experiments, with one sample referring to one eyeball from a WT or mutant mouse. Tissue sections of samples that were used for IHC were selected based on a standard set of criteria. Pupillary sections close to the middle of the eye were preferred. In addition, sections were inspected with a microscope prior staining to ensure corneal integrity, with minimally damaged, intact corneas being preferred. Sections were also assessed for general eye quality, with specific emphasis on the lens, which when fragmented could overlap with the cornea and impact staining. Only high-quality sections with intact corneas were considered for the results. For the purposes of IHC and imaging, a minimum of three sections were stained and imaged for each sample, with the highest-quality, representative sample being chosen for use in the figures included in this paper.

Quantification of cells/nuclei was conducted using the “Cell Counter” plug-in with ImageJ software. A minimum sample size of 3 was required for all cell counts, and 1 representative section was chosen for each sample. Only high-quality images of clearly stained pupillary sections were used. All images for counting were at the 20× magnification and showed the central-most aspect of the corneal epithelium. For each representative section, a count was first conducted for DAPI stained nuclei, followed by a second count for nuclei stained for the protein of interest. The percentage of stained cells was thus calculated for each sample and a mean value was determined for both the WT and mutant groups. Results were graphed, and in order to determine if any significant difference existed in the percentage of stained cells between the two groups, unpaired two-tailed *t*-tests (*p* < 0.05) were conducted (Prism9, GraphPad Software, La Jolla, CA, USA).

## 5. Conclusions

AP-2β deletion in the NCC-derived POM, as carried out in the AP-2β NCC KO model, was shown to indirectly elicit profound effects on corneal epithelial cell phenotype and stratification, the latter of which is of surface ectoderm origin. The AP-2β NCC KO cornea exhibited an absence of stratification, an expansion of K15 and loss of K12 expression. Although this keratin profile suggests conjunctival-like features, the absence of goblet cells is notable, and thus we support the alternative hypothesis, that the limbal epithelial phenotype is expanded across the entire corneal surface in the mutant. The absence of BMP4 expression in the KO epithelium may have contributed to the altered stratification, since BMP4 is a known regulator of these processes. Cross-talk between the stroma and epithelium has been shown to be critical in the development and maintenance of these tissues [6,59]. Thus, the defects we have observed in the epithelium of AP-2β NCC KO mutant mice are the result of changes to regulatory signaling from the POM-derived stroma in which AP-2β has been deleted. Our current findings, in combination with those of previous studies [8,27,37,38,45], suggest that Wnt/β-Catenin signaling in the POM-derived corneal stroma may regulate epithelial BMP4 expression, by which stratification and cell fate are influenced. Future investigation of this pathway in the AP-2β NCC KO model has the potential to improve our understanding of the underlying genetic causes that account disorders involving absence of stratification and abnormal cell phenotype in the corneal epithelium.

## Figures and Tables

**Figure 1 ijms-22-08730-f001:**
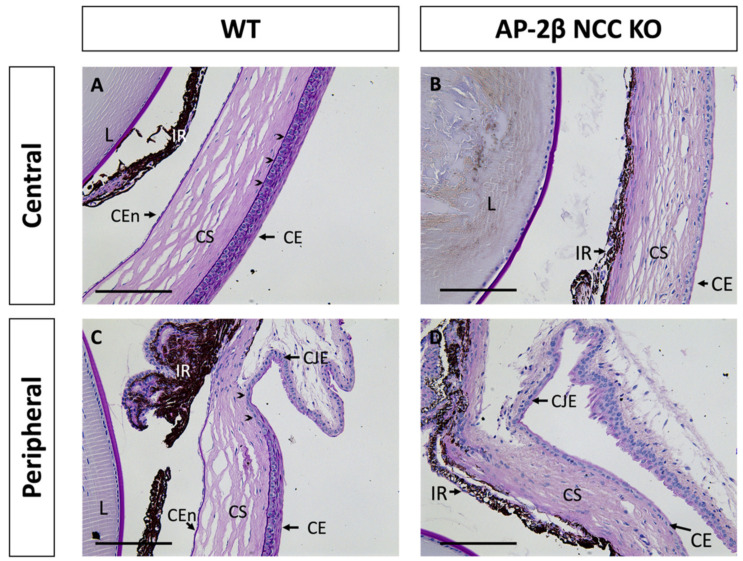
PAS staining of the anterior segment from 2–3-month-old WT and AP-2β NCC KO mice. In the WT mice (**A**,**C**), the corneal endothelium, stroma and epithelium are properly formed and distinct. A 6–7-layer stratified squamous epithelium is observed, with a dark purple stained epithelial basement membrane (arrowheads) posterior to the basal epithelial columnar cell layer (**A**). At the corneal periphery, the limbal region (arrowheads) is evident prior to conjunctiva (**C**). PAS staining of the mutant mice (**B**,**D**) displays the anticipated abnormal phenotype seen in our previous studies [16,21]. The corneal endothelium is absent, with the iris adhering to the posterior cornea, and the stroma is hypercellular relative to the WT. Epithelial stratification is absent, and cells are smaller and flatter, with staining for the basement membrane also being absent. CE, corneal epithelium; CEn, corneal endothelium; CJE, conjunctival epithelium; CS, corneal stroma; IR, iris; L, lens. Scale bars represent 150 μm.

**Figure 2 ijms-22-08730-f002:**
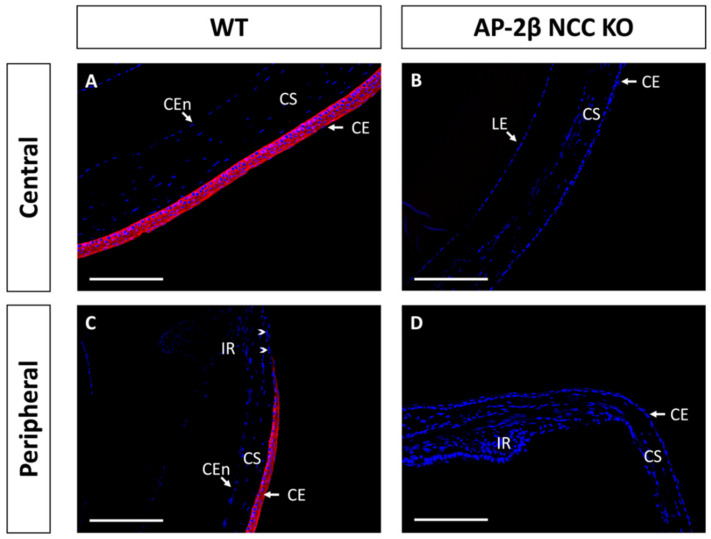
K12 IHC staining of cornea from 2–3-month-old WT and AP-2β NCC KO mice. WT mice ((**A**,**C**) *n* = 8) exhibit high expression of K12 across the central (**A**) and peripheral corneal epithelium (**C**). Peripheral K12 staining tapers off at the limbal region (arrowheads). Mature KO mice ((**B**,**D**) *n* = 8) do not express K12 in the central corneal epithelium (**B**) or at the periphery (**D**). CE, corneal epithelium; CEn, corneal endothelium; CS, corneal stroma; IR, iris; LE, lens epithelium. Scale bars represent 150 μm.

**Figure 3 ijms-22-08730-f003:**
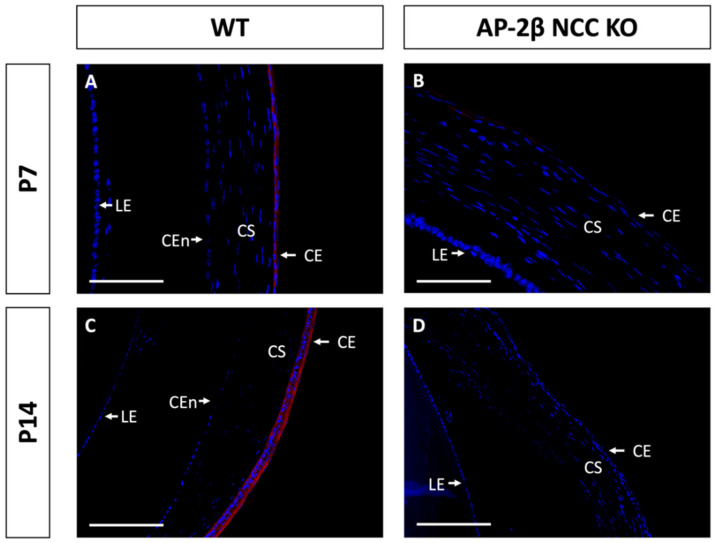
K12 staining of WT and AP-2β NCC KO mice corneas from P7 and P14. K12 is expressed consistently in the unstratified corneal epithelium of the P7 WT mouse ((**A**) *n* = 3). However, K12 is absent in the epithelium of the P7 mutant ((**B**) *n* = 4). Post-eyelid opening, at P14, K12 is highly expressed in the WT central stratified corneal epithelium ((**C**) *n* = 3). No expression of K12 is observed in the mutant epithelium at P14 ((**D**) *n* = 3). CE, corneal epithelium; CEn, corneal endothelium; CS, corneal stroma; IR, iris; LE, lens epithelium. Scale bars represent 75 μm (**A**,**B**) and 150 μm (**C**,**D**).

**Figure 4 ijms-22-08730-f004:**
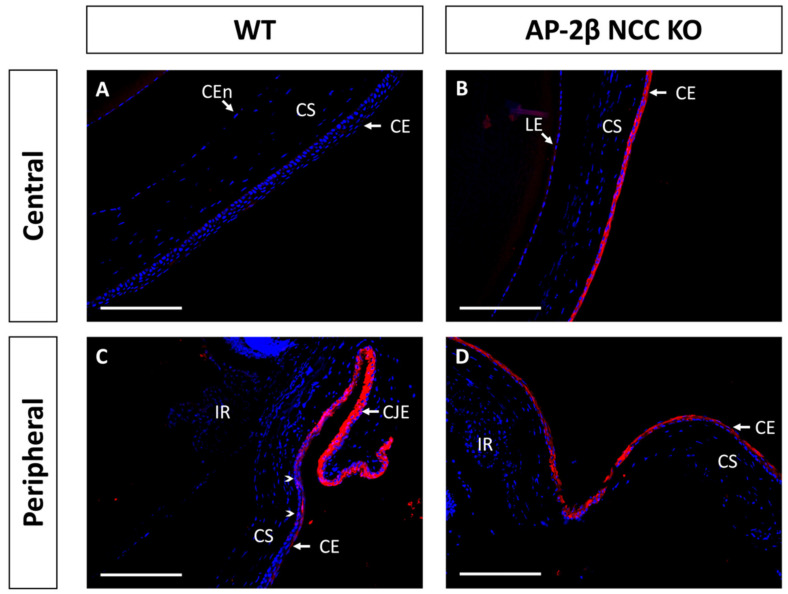
K15 IHC staining of cornea from 2–3-month-old WT and AP-2β NCC KO mice. In the WT ((**A**,**C**) *n* = 8) expression of K15 is limited to the conjunctival and limbal epithelia (**C**). K15 staining tapers off at the transition to the peripheral corneal region (arrowheads) with the corneal epithelium not staining for K15 (**A**). In contrast, for the mutant ((**B**,**D**) *n* = 7), K15 is observed at high levels across the central (**B**) and peripheral cornea (**D**), continuous with the staining observed at the limbus and conjunctival epithelia. CE, corneal epithelium; CEn, corneal endothelium; CJE, conjunctival epithelium; CS, corneal stroma; IR, iris; LE, lens epithelium. Scale bars represent 150 μm.

**Figure 5 ijms-22-08730-f005:**
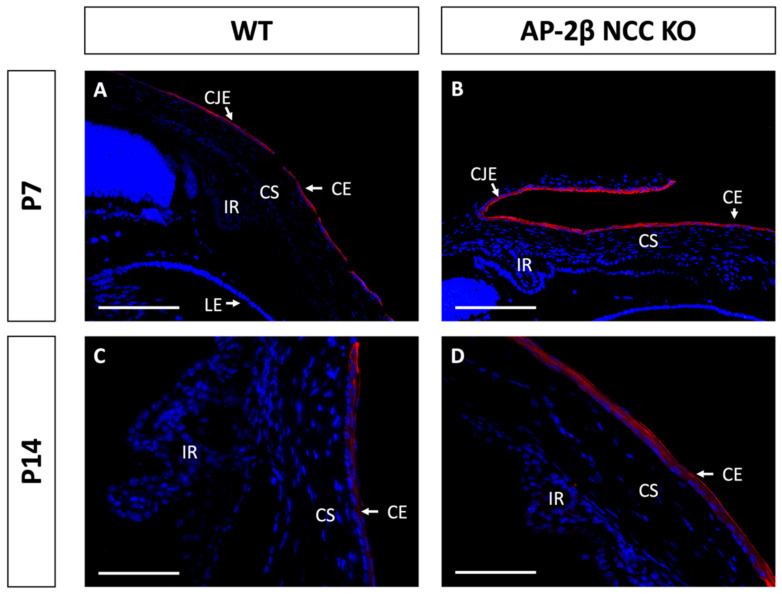
K15 staining of WT and AP-2β NCC KO mice corneas from P7 and P14. K15 is observed to be expressed continuously across epithelium of the conjunctiva, limbus and cornea of P7 WT mice ((**A**) *n* = 3); this same expression pattern is seen for the mutant at the P7 timepoint ((**B**); *n* = 4). At P14 for the WT cornea, we see that K15 expression is present in the limbus. However, it tapers off approaching the peripheral corneal epithelium ((**C**) *n* = 4). In contrast, the mutant continues to exhibit continuous expression of K15 across all of the conjunctival, limbal and corneal epithelia ((**D**) *n* = 3). CE, corneal epithelium; CEn, corneal endothelium; CS, corneal stroma; IR, iris; LE, lens epithelium. Scale bars represent 150 μm (**A**,**B**) and 75 μm (**C**,**D**).

**Figure 6 ijms-22-08730-f006:**
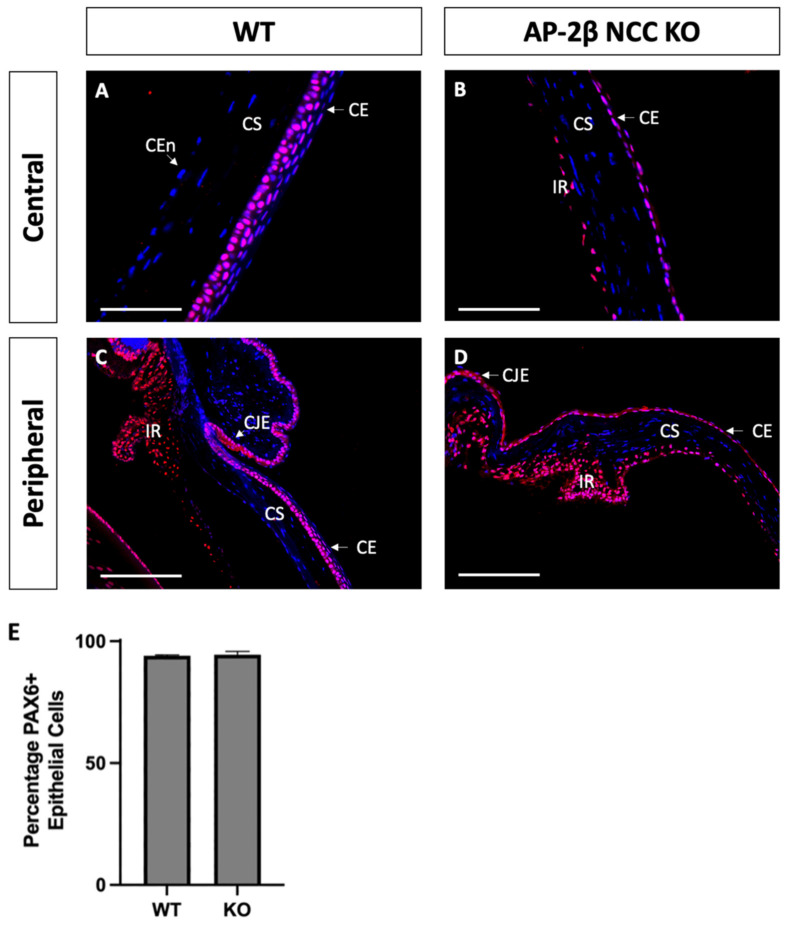
PAX6 IHC staining of cornea from 2–3-month-old WT and AP-2β NCC KO mice. Nuclear PAX6 staining is observed in WT mice ((**A**,**C**) *n* = 4), at consistent levels both in the central region (**A**) and the periphery (**C**), continuous with the expression at the limbus and conjunctiva. PAX6 expression in the mutant ((**B**,**D**) *n* = 3) was comparable to the WT, with the central corneal (**B**), peripheral corneal (**D**), limbal and conjunctival regions showing a similar proportion of nuclear cell staining. Cell counts indicated no significant different (*p* > 0.05) in the percentage of PAX6-positive cells between the WT (94.03% ± 0.38; *n* = 3) and mutant (94.44% ± 1.42; *n* = 3) (**E**). Error bars represent standard deviation. CE, corneal epithelium; CEn, corneal endothelium; CJE, conjunctival epithelium; CS, corneal stroma; IR, iris; LE, lens epithelium. Scale bars represent 75 μm (**A**,**B**) and 150 μm (**C**,**D**).

**Figure 7 ijms-22-08730-f007:**
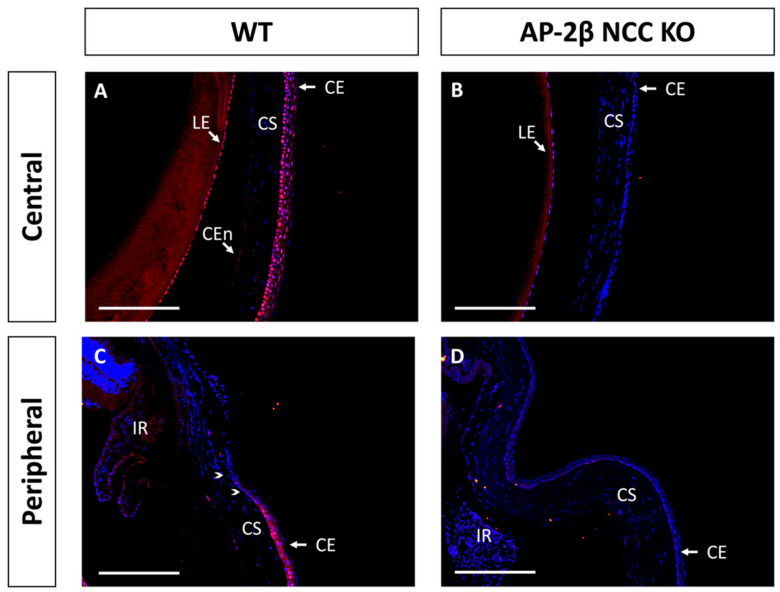
BMP4 IHC staining of cornea from 2–3-month-old WT and AP-2β NCC KO mice. WT mice ((**A**,**C**) *n* = 5) display nuclear epithelial expression of BMP4 across the central corneal epithelium (**A**), with the most consistent expression observed for the basal layer. In the peripheral (**C**) corneal epithelium, we continue to see this pattern of nuclear staining. However, BMP4 expression is absent from the limbal (arrowheads) and conjunctival epithelial regions. BMP4 expression is not conserved in the mutant (**B**,**D**) *n* = 5) with staining being absent from the central (**B**) and peripheral regions (**D**). CE, corneal epithelium; CEn, corneal endothelium; CS, corneal stroma; IR, iris; LE, lens epithelium. Scale bars represent 150 μm.

**Figure 8 ijms-22-08730-f008:**
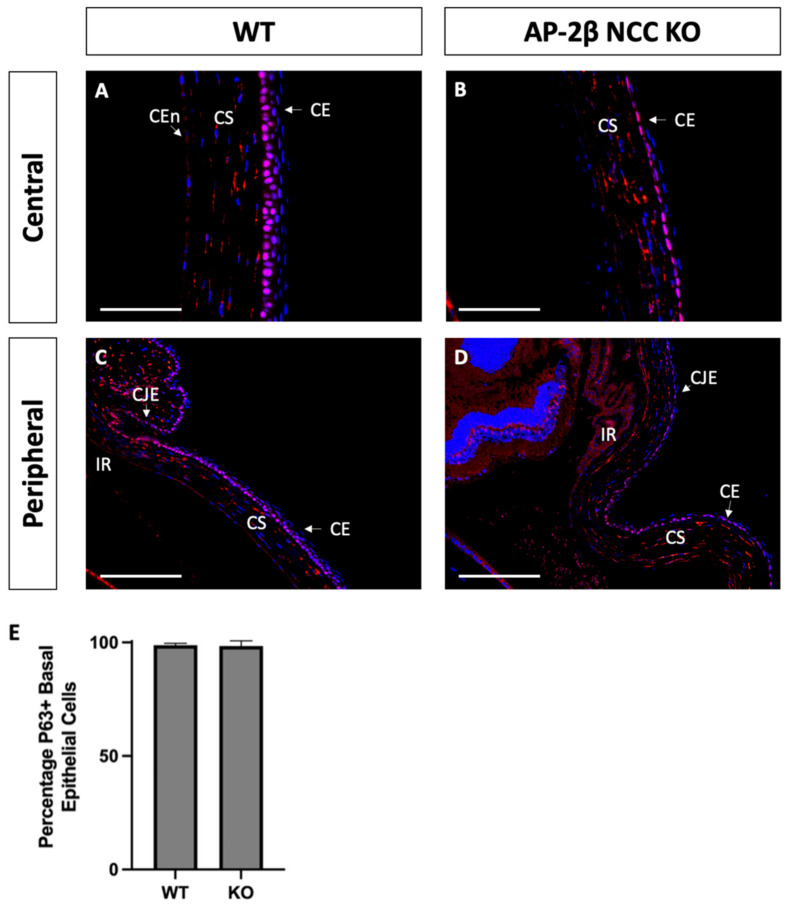
P63 IHC staining of cornea from 2–3-month-old WT and AP-2β NCC KO mice. P63 in WT mice ((**A**,**C**) *n* = 4) is observed to be expressed consistently across the basal layer of the corneal epithelium at both the central (**A**) and peripheral (**C**) regions. This staining continues throughout the limbus and conjunctiva without any significant different in the proportion of cells stained relative to the differentiated corneal epithelium. Consistent stromal expression of P63 is also observed throughout the entire extent of the cornea. The mutant cornea ((**B**,**D**) *n* = 4) showed the same pattern of P63 expression as the WT, with the central corneal epithelium (**B**) displaying nuclear P63, continuous with that staining observed in the epithelial periphery, limbus and conjunctiva (**D**). Cell counts indicated no significant different (*p* > 0.05) in the percentage of P63-positive cells between the WT (98.04% ± 0.90; *n* = 4) and mutant (95.16% ± 2.28; *n* = 4) (**E**). Error bars represent standard deviation. Stromal expression throughout the cornea is also similar to that observed in the WT. CE, corneal epithelium; CEn, corneal endothelium; CJE, conjunctival epithelium; CS, corneal stroma; IR, iris; LE, lens epithelium. Scale bars represent 75 μm (**A**,**B**) and 150 μm (**C**,**D**).

**Figure 9 ijms-22-08730-f009:**
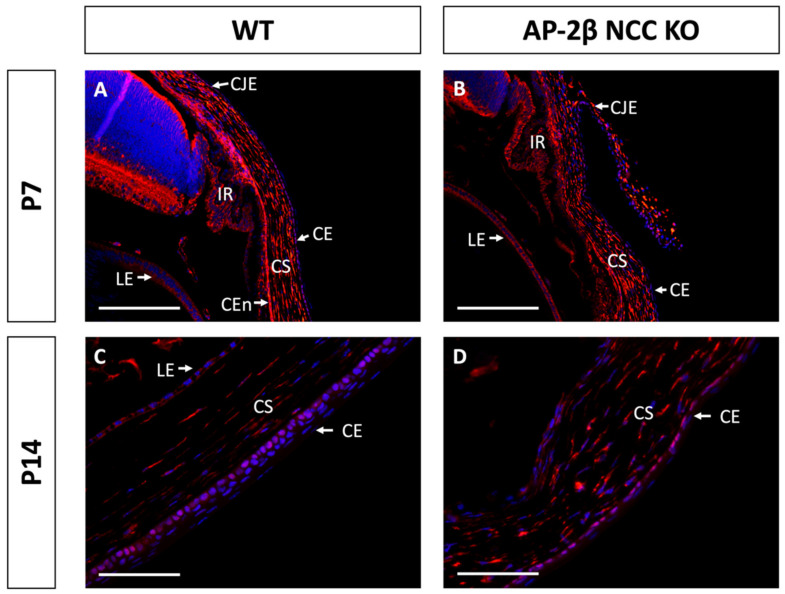
P63 staining of WT and AP-2β NCC KO mice corneas from P7 and P14. The pattern of P63 expression is similar between the WT ((**A**) *n* = 4) and mutant ((**B**) *n* = 4) at P7, with high stromal keratocyte expression as well as consistent nuclear staining of the corneal, limbal and conjunctival epithelia. In the WT at P14, initial stratification is observed and P63 expression remains confined to the basal cell nuclei of the corneal epithelium ((**C**) *n* = 3). In the KO, though stratification is not seen, nuclear basal epithelial cell expression of P63 is consistent across the epithelium ((**D**) *n* = 3). Stromal expression of P63 is similar for the WT and mutant at P14. CE, corneal epithelium; CEn, corneal endothelium; CJE, conjunctival epithelium; CS, corneal stroma; IR, iris; LE, lens epithelium. Scale bars represent 150 μm (**A**,**B**) and 75 μm (**C**,**D**).

## Data Availability

The data presented in this study are available in this article.

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
