# Peer review of "Conditional Deletion of AP-2β in the Periocular Mesenchyme of Mice Alters Corneal Epithelial Cell Fate and Stratification"

_ijms, 2021, doi:10.3390/ijms22168730_

Round 1
Reviewer 1 Report
In the abstract, purpose, methods and materials, results, and conclusions are appropriately stated. The content presented in the Discussion section is appropriate and does not repeat information from the Results section.
Minor points
- Author observed that the corneal endothelium is absent, with the iris adhering to the posterior cornea, and the stroma is hypercellular relative to the WT (Figure 1, line 107-108). A break in functional integrity of the corneal endothelium associated with corneal edema. How about the trend of corneal thickness in AP-2 NCC KO mice.
- How about the PAX6 IHC staining of lens epithelium from 2-3-month-old WT and AP-2NCC KO mice (Figure 4). Please describe it in Discussion, even if it is an excerpt from the reported cases.
Author Response
We thank the reviewers for their comments and are pleased that they found our manuscript of value to the field, providing new clues to the process of corneal development and epithelial specification. We have performed analyses suggested by the reviewers and updated the text as outlined below. We have also addressed the reviewers’ concerns and responded to the comments in point-by-point fashion.
Reviewer 1
1. Author observed that the corneal endothelium is absent, with the iris adhering to the posterior cornea, and the stroma is hypercellular relative to the WT (Figure 1, line 107-108). A break in functional integrity of the corneal endothelium associated with corneal edema. How about the trend of corneal thickness in AP-2b NCC KO mice?
To answer the reviewer’s question above, we have now included relevant information in our introduction section when introducing the model (P2, 66-68). A decrease in corneal epithelial thickness of AP-2b NCC KO mice was observed when compared to the control due to the fact that the corneal epithelial cell layer did not stratify. We did not measure overall thickness of the cornea, however we reported in a previous publication that “the mutant cornea was less compact, with large gaps within the stroma” (Martino et al, 2016). In the same publication, OCT images of the cornea were shown that demonstrate that the thickness of the stroma is relatively the same between the mutant and WT.
2. How about the PAX6 IHC staining of lens epithelium from 2-3-month-old WT and AP-2b NCC KO mice (Figure 4). Please describe it in Discussion, even if it is an excerpt from the reported cases.
We thank the reviewer for bringing this point to our notice. We have now included a description of the PAX6 IHC staining pattern in lens epithelium for the WT and AP-2b NCC KO mice in the results (P6, 188-190). As expected, we did not observe any change in the PAX6 staining pattern in AP-2b NCC KO lens epithelium when compared to WT lens epithelium as the lens is derived from surface ectoderm and was therefore not targeted by the Wnt1Cre transgene.
Reviewer 2 Report
In the manuscript entitled: “Conditional Deletion of AP-2b in the Periocular Mesenchyme of Mice alters Corneal Epithelial Cell Fate and Stratification” the authors use a genetic strategy to selectively inactivate AP-2B in the NCC population of cells and subsequently analyze corneal phenotypes at developmental and adult stages. In the absence of AP-2b in the NCC population the authors observe structural and cell fate changes in the corneal epithelium and propose that this results from the inability to properly signal from the POM-derived stroma. Mechanistically they suggest reduced BMP4 expression may be a driver of these phenotypes.
This work is a follow up to a recent publication outlining the consequences of AP-2b inactivation in POM on the retina in order to follow up on observed anterior segment phenotypes. Overall, the manuscript is of value to the field and provides new clues to the process of corneal development and epithelial specification. In its current state there are a few points that should be addressed, in particular the aspect of scientific rigor and quantification.
Major points:
- For all of the figures there is a general lack of the degree of scientific rigor. How many different sections were imaged? The authors often use terms like “significantly different” and “similar proportion” without producing any statistical/quantified data. Was normalization between WT and mutants taken into account, especially if the mutant eyes were mis-shaped or smaller? In order for the results to be impactful, quantification of the results should be included and only after statistical analysis can significance be reported.
- There is a lack of detailed methods on how the authors ensured that similar regions of the eyes/AS was collected for their sections, especially if the mutants have major physiological defects.
- The authors include some nice data from developmental stages, P7, P14 which is currently in the supplementary material. I would suggest this data be included in the main figures as it not only supports their main points but also adds a new dimension to the work by also tracking expression patterns over developmental time.
Minor points:
- Figure 1 could be more informative with the inclusion of what the whole eyes of the WT vs mutants look like.
- Line 122-125, the sentence here reads rather awkwardly, I suggest re wording it to be clearer.
- Line 146-148: again, the sentence here reads awkwardly.
- Figure 4: the associated text in the MS suggests there is no difference in Pax6 staining, yet its clear there are more cells labelled in the WT CE (A) than in the mutant CE (B)? IN addition, the IR appears expanded in the mutant (D). This may be an issue pertaining to normalization/quantification of the data but it should be addressed.
- Figure 6: Again the authors suggest there is no difference between WT (A) and mutant (B) for p63 staining, yet in the image there is clearly fewer labeled cells in the mutant?
Author Response
We thank the reviewers for their comments and are pleased that they found our manuscript of value to the field, providing new clues to the process of corneal development and epithelial specification. We have performed analyses suggested by the reviewers and updated the text as outlined below. We have also addressed the reviewers’ concerns and responded to the comments in point-by-point fashion.
Major points:
- For all of the figures there is a general lack of the degree of scientific rigor. How many different sections were imaged? The authors often use terms like “significantly different” and “similar proportion” without producing any statistical/quantified data. Was normalization between WT and mutants taken into account, especially if the mutant eyes were mis-shaped or smaller? In order for the results to be impactful, quantification of the results should be included and only after statistical analysis can significance be reported.
As suggested by the reviewer, we have now performed statistical analyses for PAX6 and P63 IHC staining to support our findings for statements of significance. Graphs comparing the number corneal epithelial cells expressing PAX6 and P63, in the central cornea have now been included and the respective figure captions and discussion has been modified. (P7, F6, 199-201) (P9, F8 252-255) (P12, 374). In the context of BMP4, we observed either a presence or a complete absence of BMP4 staining in the WT or mutant, respectively. Since this is a complete on/off result for all the samples evaluated (n=5 eyes), we have changed our description to use the term “completely altered expression pattern” in the text. With regard to scientific rigor, additional detail was added to the methods describing the use of samples and specific sections in reference to specimen quality for selection and use in analysis (P14, 466-487).
- There is a lack of detailed methods on how the authors ensured that similar regions of the eyes/AS was collected for their sections, especially if the mutants have major physiological defects.
We have now included the criteria for sample/section selection, including ensuring use of sections from similar regions of eye/AS, in our methods section (P14, 466-476). The pupil was considered as a reference point for both mutant and WT eyes and similar sections were chosen for further staining – more detail is provided in the methods.
- The authors include some nice data from developmental stages, P7, P14 which is currently in the supplementary material. I would suggest this data be included in the main figures as it not only supports their main points but also adds a new dimension to the work by also tracking expression patterns over developmental time.
As suggested by the reviewer, we have now included some images from P7 and P14 developmental stages as main figures. This has been done for K12, K15 and P63 staining. The text has also been modified accordingly. Please refer to Figure 3 and captions, as well as text on page 3 line 124 to 127. Also, please see Figure 5 and captions, as well as text on page 5 line 151 to 159. Finally, Figure 9 and captions, as well as text on page 8 lines 239 to 241.
Minor points:
- Figure 1 could be more informative with the inclusion of what the whole eyes of the WT vs mutants look like.
In a previous paper we showed low power images of whole eyes in order to demonstrate the closed angle phenotype of the anterior segment of the eye (Martino et al., 2016). These images are beyond the scope of this study which only addresses the cornea, and predominantly the corneal epithelium.
- Line 122-125, the sentence here reads rather awkwardly, I suggest re wording it to be clearer.
As suggested by the reviewer, we have now reworded the sentence (line 124-125).
- Line 146-148: again, the sentence here reads awkwardly.
As suggested by the reviewer, we have now reworded the sentence (line 148).
- Figure 4: the associated text in the MS suggests there is no difference in Pax6 staining, yet it is clear there are more cells labelled in the WT CE (A) than in the mutant CE (B)? IN addition, the IR appears expanded in the mutant (D). This may be an issue pertaining to normalization/quantification of the data but it should be addressed.
We agree with the reviewer that in the previously submitted figures for the paper, PAX6 appeared to be reduced in the mutant epithelium relative to the WT, and this was due to the fact at the magnification shown it was difficult to observe the expression of PAX6 in the basal layer of the mutant epithelium. In order to approach this, two changes were made. First, the two central images (A and B) for PAX6 in Figure 6 have been replaced with higher magnification images which better show the staining. In addition, we have now performed cell counts and statistical analyses for PAX6 stained cells (T-test). The graph and the modified text have been included in the manuscript (P7, F6, E, 199-201). We believe the IR appears expanded in the mutant due to its adhesion to the cornea. In the WT, the angle is open and the iris appears away from the corneal endothelium that further makes it appear smaller in dimension when compared to adhered iris in the mutant. The protocol used for selecting samples/section, and cell counting as well as analysis has been added to the methods section (P14, 466-487).
- Figure 6: Again the authors suggest there is no difference between WT (A) and mutant (B) for p63 staining, yet in the image there is clearly fewer labeled cells in the mutant?
We agree with the reviewer that in the previously submitted images for the paper, P63 does appear to be reduced in the mutant basal epithelium relative to the WT. In order to approach this, two changes were made. First, the two central images (A and B) for P63 in Figure 9 have been replaced with higher magnification images which better show the overlapping staining in the mutant basal corneal epithelium particularly. In addition, we have now performed cell counts and statistical analyses (T-test) for P63 stained cells (P9, F8, E, 252-255). The graph and the modified text have been included in the manuscript. The protocol used for selecting samples/sections, and cell counting as well as analysis has been added to the methods section (P14, 466-487).
Round 2
Reviewer 2 Report
The authors have satisfied my concerns and I appreciate their efforts. I therefore recommend this work for publication.